# Thrombophilia Impact on Treatment Decisions, Subsequent Venous or Arterial Thrombosis and Pregnancy-Related Morbidity: A Retrospective Single-Center Cohort Study

**DOI:** 10.3390/jcm11144188

**Published:** 2022-07-19

**Authors:** Kristina Vrotniakaite-Bajerciene, Tobias Tritschler, Katarzyna Aleksandra Jalowiec, Helen Broughton, Justine Brodard, Naomi Azur Porret, Alan Haynes, Alicia Rovo, Johanna Anna Kremer Hovinga, Drahomir Aujesky, Anne Angelillo-Scherrer

**Affiliations:** 1Department of Hematology and Central Hematology Laboratory, Bern University Hospital, 3010 Bern, Switzerland; katarzynaaleksandra.jalowiec@insel.ch (K.A.J.); helen.broughton@students.unibe.ch (H.B.); justine.brodard@insel.ch (J.B.); naomiazur.porret@insel.ch (N.A.P.); alicia.rovo@insel.ch (A.R.); johanna.kremer@insel.ch (J.A.K.H.); anne.angelillo-scherrer@insel.ch (A.A.-S.); 2Department for BioMedical Research, University of Bern, 3008 Bern, Switzerland; 3Department of General Internal Medicine, Bern University Hospital, 3010 Bern, Switzerland; tobias.tritschler@insel.ch (T.T.); drahomirantonin.aujesky@insel.ch (D.A.); 4Clinical Trials Unit Bern, University of Bern, 3012 Bern, Switzerland; alan.haynes@ctu.unibe.ch

**Keywords:** thrombophilia, venous thrombosis, arterial thrombosis, pregnancy-related morbidity, clinical decision-making

## Abstract

(1) Background: Thrombophilia testing utility has remained controversial since its clinical introduction, because data on its influence on treatment decisions are limited. (2) Methods: We conducted a single-center retrospective cohort study of 3550 unselected patients referred for thrombophilia consultation at the Bern University Hospital in Switzerland from January 2010 to October 2020. We studied the influence of thrombophilia testing results on treatment decisions and evaluated the association between thrombophilia and thromboembolic and pregnancy-related morbidity events after testing up to 03/2021. (3) Results: In 1192/3550 patients (34%), at least one case of thrombophilia was found and 366 (10%) had high-risk thrombophilia. A total of 211/3550 (6%) work-ups (111/826 (13%) with low-risk thrombophilia and 100/366 (27%) with high-risk thrombophilia) led to an appropriate decision to extend or initiate anticoagulation, and 189 (5%) negative results led to the withholding of anticoagulation therapy inappropriately. A total of 2492 patients (69%) were followed up for >30 days, with a median follow-up of 49 months (range, 1–183 months). Patients with high-risk thrombophilia had a higher risk of subsequent venous thromboembolic events and pregnancy-related morbidity compared to those without thrombophilia. (4) Conclusions: Our study demonstrated the limited usefulness of thrombophilia work-up in clinical decision-making. High-risk thrombophilia was associated with subsequent venous thromboembolism and pregnancy-related morbidity.

## 1. Introduction

The clinical utility of thrombophilia testing has remained a subject of controversy since its introduction in clinical practice [1]. As guidelines of thrombophilia testing include only conditional recommendations, patterns of thrombophilia testing vary strongly across centers [2,3,4,5]. Considering the high cost of the work-up, testing is dependent on patients’ and physicians’ individual preference, financial status and the local healthcare system [6]. Moreover, the influence of test results on treatment decisions is still a matter of ongoing debate [1,4].

Data showing the clinical usefulness and benefits of positive thrombophilia testing results for further clinical decisions on treatment after venous thromboembolism (VTE) are limited. Garcia-Horton et al. showed the limited relevance of thrombophilia work-up in clinical decision-making after unprovoked VTE at a tertiary thrombosis center in Canada [7]. Given the growing body of evidence of the rather limited impact of thrombophilia upon the recurrence of VTE and overall mortality [8,9,10,11,12], current scientific data emphasize strict but variable selection criteria for thrombophilia testing [5,13,14,15,16,17].

Since most studies on thrombophilia testing focus on accuracy and guideline interpretation [18,19,20], the quantification of its guidance on treatment decisions and impact on VTE outcome remains unclear. The potential negative effect of withholding anticoagulation or overtreating patients because of the work-up is being debated, whereas its significance is uncertain [1]. Thrombophilia work-up and its clinical consequences in patients with arterial thrombosis, pregnancy-related morbidities or asymptomatic thrombophilia carriers is even less defined [21,22,23,24] and clinical trials in this context are absent [25]. 

To investigate the impact of thrombophilia testing on the management and clinical course of thromboembolic disease and pregnancy-related morbidity, we conducted a large 10-year single-center retrospective cohort study. We analyzed the work-up patterns of thrombophilia and their impact on treatment decisions, and the subsequent occurrence of venous and arterial thrombosis and pregnancy-related morbidities. 

## 2. Materials and Methods

### 2.1. Study Design and Patients

A single-center retrospective cohort study was conducted at the Department of Hematology of the Bern University Hospital in Switzerland between January 2010 and October 2020. We systematically screened consecutive patients referred for testing of hereditary and/or acquired thrombophilia by general practitioners or non-hematologist medical specialists with the support of the hospital data management service using internal specified codes for thrombophilia work-up. Patients with general consent were included if thrombophilia testing was performed and they had a documented history of objectively confirmed VTE and/or arterial thrombosis in any location, a history of pregnancy-related morbidity or were referred for thrombophilia testing due to a positive family history for VTE or hereditary thrombophilia. 

The objective diagnosis of deep vein thrombosis (DVT), superficial vein thrombosis and muscle vein thrombosis was defined by a positive compression ultrasonography or venography [26], and pulmonary embolism (PE) was defined by a new high-probability ventilation/perfusion lung scan or a new contrast filling defect on spiral computed tomography (CT) or pulmonary angiography [26]. Arterial thrombosis was defined by the presence of stroke on brain magnetic resonance (MR) imaging or CT or by a diagnostic coronary angiography in patients with myocardial infarction [27,28]. A critical limb ischemia of a peripheral artery disease was established by arterial Doppler ultrasound, CT angiography, MR angiography or catheter-based arteriography [29]. Renal artery or vein thrombosis, splanchnic vein thrombosis, cerebral vein thrombosis and thrombosis of aorta and vena cava were defined as filling defects in the corresponding vessel on CT or MR angiography or venography [30,31,32]. Other types of thromboembolism, such as retinal vein or artery thrombosis, penis vein thrombosis, osteonecrosis or chronic inflammatory disease, along with pregnancy-related morbidity, were defined by relevant specialists using a referral report. Pregnancy-related morbidities were defined as pregnancy loss at all gestational ages, placenta failure, preeclampsia [33] and HELLP syndrome (hemolysis, elevated liver enzyme levels, low platelet count) according to obstetrical diagnostic criteria [34]. 

Clinical data were collected from structured electronic forms using a standardized case report form and entered into a computerized database (REDCap software) by two persons. Data comprised demographic characteristics of patients and their family history of VTE in first- and second-degree relatives, details of all previous thrombotic events or pregnancy-related morbidity (date and location), risk factors for most recent VTE and arterial thrombosis event (namely heavy smoking (>20 pack years), immobilization > 4 h, infections requiring bedrest > 3 days, estrogen-based medications, pregnancy and peripartum, intravenous catheters, active cancer, obesity (body mass index [BMI] > 30 kg m^−2^), trauma, surgery, cancer medication, presence of extended varicose veins, patent foramen ovale or other septal defect) and co-morbidities (diabetes mellitus, arterial hypertension, liver cirrhosis, kidney failure, rheumatic disease, depression, chronic inflammatory disease, dyslipidemia, cardiovascular diseases, pulmonary diseases, neurological diseases).

### 2.2. Thrombophilia Testing

Thrombophilia testing was performed between 3 and 6 months following the index event, after the evaluation of the patient by a hematologist, taking into consideration age, risk factors, family history of VTE, co-morbidities and type of thrombosis or pregnancy-related morbidity. A thrombophilia work-up was considered as “performed” if one or more of the following thrombophilia parameters were tested: factor V Leiden (FVL) mutation status, prothrombin gene 20210G>A mutation status, protein C (PC) and antithrombin (AT) activity as well as free protein S (PS) antigen, lupus anticoagulant (LA), anticardiolipin antibodies and anti-β2-glycoprotein I antibodies. Only results of accurate thrombophilia testing were considered, excluding PC and PS testing whilst on vitamin K antagonists (VKA) or PS level during pregnancy.

Testing for PC (Protein C Berichrom^®^, Siemens, Marburg, Germany; Protein C COAG, Siemens, Marburg, Germany), PS (Free protein S, Asserachrom^®^, Diagnostica Stago, Asnières, France from 2010 to 2015,; Free Protein S Antigen, Innovance^®^, Siemens, Marburg, Germany from 2015 to 2020) and AT activity (LR Antithrombin, Coamatic^®^, Diapharma, Bedford, USA from 2010 to 2013; LRT Antithrombin, Biophen^®^, Endotell, Allschwil, Switzerland from 2013 to 2014, and Antithrombin Innovance^®^, Siemens, Marburg, Germany from 2014 to 2020) was performed in the routine hemostasis laboratory (Bern University Hospital). Antiphospholipid antibodies were tested using Varelisa diagnostic kits (Phadia^®^, ThermoFisher, Freiburg, Germany) from 2010 to 2014, fluorescence enzyme immunoassay (Phadia^®^ 250, ThermoFisher, Freiburg, Germany) from 2014 to 2015 and automated chemiluminescence assay (Bio-flash^®^, Inova Diagnostics, San Diego, USA) from 2015 to 2020 and dilute Russell’s viper venom time (Cryocheck^®^, Endotell, Allschwil, Switzerland). The diagnosis of an antiphospholipid antibody syndrome was established by persistent laboratory evidence of antiphospholipid antibodies at least 12 weeks later and the presence of VTE, arterial thrombosis or criteria pregnancy-related morbidity [35]. Genetic mutations were detected by the polymerase chain reaction method (FVL and Prothrombin, RealFast Assay^®^, Vienna Lab Diagnostics, Vienna, Austria). 

### 2.3. Classification of Thrombophilia and Risk Factors

Categorization as minor and major provoking risk factors was based on the guidance provided by the International Society on Thrombosis and Haemostasis (ISTH) [36]. In addition to the ISTH-based criteria, the presence of an intravenous catheter [37] and May-Thurner syndrome (>70% iliofemoral compression) [38] were categorized as major risk factors, whereas immobilization > 4 h [39] and heavy smoking (>20 pack years) [40] as minor risk factors. VTE in the presence of merely an environmental risk factor (male sex and older age) was categorized as unprovoked thromboembolism.

Minor thrombophilia was defined as isolated heterozygous FVL or prothrombin 20210G>A mutation according to institutional guidelines. AT activity < 70%, PC activity < 69% and PS free antigen < 59%, antiphospholipid antibody syndrome, homozygous FVL or prothrombin 20210G>A or any compound thrombophilias were considered as high-risk thrombophilia.

### 2.4. Follow-Up and Outcomes

The primary study outcome was the influence of thrombophilia testing on anticoagulation management decisions. Secondary outcomes included the occurrence of first or recurrent VTE, arterial thrombosis or pregnancy-related morbidity after thrombophilia testing. 

The influence of thrombophilia testing on management decisions was assessed from a structured medical report of each patient by two persons and classified as follows: (1) no influence on management; (2) appropriate management decision and (3) results not considered/overlooked, meaning a positive or potential positive influence; (4) decision to overtreat and (5) decision to undertreat, showing a negative influence (Table 1). Because of the implementation of a structured reporting form at our center before 2010, considering clinical and laboratory factors leading to the choice of management, we were able to document clinical decisions on treatment that were merely based on thrombophilia. Therefore, management decisions based on clinical characteristics of thrombotic event (e.g., unprovoked or recurrent VTE), patient family history for VTE, preferences or high-bleeding risk were categorized as non-influential, irrespective of the thrombophilia result. We exclusively assessed the thrombophilia-based management decisions on prophylactic and therapeutic anticoagulation treatment, and did not consider other decisions or patient education, such as avoidance of estrogen-based treatment, change in type of anticoagulant or lifestyle modification. Guidance for the appropriateness of management decisions regarding thrombotic events, pregnancy-related morbidity or asymptomatic carriership of thrombophilia was based on international guidelines [16,17,41,42]. 

Data from all complete hospital records, including other disciplines, for the identification of subsequent VTE, arterial thrombosis or pregnancy-related morbidity after the thrombophilia consultation were investigated until March 2021. Only objectively confirmed events, according to the previously mentioned inclusion criteria, were considered. As 36 patients had a partial thrombophilia work-up before January 2010 in our clinic and were referred for a second time, the start of the follow-up time was defined by the first performed work-up from December 2004 to March 2021. For time-to-event analyses, we considered only patients with a follow-up >30 days and censored patients at the time of last hospital record or time of event. 

### 2.5. Statistical Analysis 

Continuous variables are presented as mean ± standard deviation (SD) and were compared using one-way ANOVA. Categorical variables are shown as percentages and compared with x^2^ test. Univariable logistic regression models were used to assess the influence of thrombophilia on treatment decisions. As the negative result did not lead to any influence on treatment, heterozygous FVL mutation was used as a reference due to its lowest impact on treatment decision. Associations between low- and high-risk thrombophilia and the time to new thrombotic event or pregnancy-related morbidity were assessed using Cox proportional hazard models, yielding hazard ratios (HR) with their corresponding 95% confidence intervals (CI) and *p*-values for the failure event of primary interest. We adjusted the models for previously published predictors of venous and arterial thrombosis, including age > 50 years, male sex and risk factors and co-morbidities, such as smoking, diabetes mellitus, obesity, arterial hypertension, kidney failure, dyslipidemia, depression, chronic inflammatory disease and active cancer [43,44,45]. The Kaplan–Meier method was used to plot time from work-up for thrombophilia to recurrence of new thrombotic event or pregnancy-related morbidity. Only complete case analysis was performed, without an attempt to replace missing values with imputation methods. A value of *p* < 0.05 was considered statistically significant. All analyses were performed with R 4.1.1 and figures were edited with GraphPad Prism v9.1.2.

## 3. Results

### 3.1. Study Cohort 

Of 5064 patients screened for eligibility, we excluded 1356 patients (27%) without general consent, 136 patients (3%) because no thrombophilia work-up was performed and 22 patients (0.4%) because of the absence of the objective documentation of thromboembolism or pregnancy-related morbidity, leaving a study sample of 3550 patients (Figure 1). Of them, 2429 patients (68%) had a follow-up of more than 30 days and were considered for time-to-event analysis. Their median follow-up duration was 49 months (range, 1–183 months).

At the time of thrombophilia work-up, the mean age was 42 years (±15) and 2118 patients (60%) were women (Table 2). Most patients (2343, 66%) were referred because of VTE, mainly DVT and/or PE (1791/2343, 76%), whereas 583 (16%) patients had a positive history of unexplained arterial thrombosis, mainly stroke (444/583, 76%) (Appendix A). A total of 504 (14%) patients had no prior thromboembolic event, but a positive family history for VTE in first-degree (306/504, 61%) or second-degree (143/504, 28%) relatives. A minority of referrals were due to pregnancy-related morbidity (120, 3%). Most patients (1999, 56%) had no documented co-morbidities and one third (1259, 35%) had no documented risk factors for VTE or arterial thrombosis (Table 2). Co-morbidities and risk factors of the cohort patients are presented in Appendix A, comprising mostly arterial hypertension (578, 16%) and dyslipidemia (405, 11%) as risk factors for arterial thrombosis, and immobilization > 4 h (743, 21%) and estrogen-based medication (706, 20%) representing minor risk factors for VTE. The major risk factors for VTE were found in 415 (12%) cohort patients (Table 2). 

### 3.2. Prevalence of Thrombophilia in the Cohort Study

A total of 1260 thrombophilias were found in 1192 (34%) patients. The most common type of thrombophilia was heterozygous FVL mutation (714 patients, 20%), followed by heterozygous prothrombin 20210G>A mutation (193 patients, 5%) and antiphospholipid antibody syndrome (119 patients, 3%) (Figure 2). One hundred and seven patients (3%) had more than one thrombophilia. Patients with thrombophilia were younger and were less likely to have co-morbidities or risk factors. More patients with thrombophilia were referred because of a recurrent VTE. Patients with arterial thrombosis had significantly less positive thrombophilia results compared to other patient groups, whereas women with pregnancy-related morbidities and asymptomatic patients with a positive family history were more likely to test positive (Table 2). 

### 3.3. Impact of Thrombophilia Testing on Treatment Decisions

In 3050 patients (86%), a thrombophilia work-up did not have any influence on the treatment decision, mostly when patients tested negative (2171 patients, 71%) or confirming low-risk thrombophilia (671 patients, 22%) (Table 3). A total of 211 positive work-ups (6%) led to an appropriate decision to extend or initiate anticoagulation; 82 positive work-ups (2.2%) were inappropriately overlooked—21 (26%) with antiphospholipid antibody syndrome, 20 (24%) with heterozygous factor V Leiden mutation and 14 (17%) with heterozygous prothrombin 20210G>A mutation. Of 195 patients (5.4%) with an inappropriate treatment decision, 181 patients (93%) had a negative thrombophilia work-up. Only 11 patients (0.3%) had a positive work-up that led to overtreatment. 

The presence of antiphospholipid antibody syndrome had the highest positive influence on treatment decision (71/119, 60%), followed by high-risk hereditary thrombophilia (86/247, 35%) (Table 3). However, only 17% positive results (136/826) for low-risk thrombophilia led to a change in treatment, and merely 116 out of 714 (16%) carrierships of a heterozygous FVL mutation provided further guidance (Appendix A). Compared with the presence of a heterozygous FVL mutation, antiphospholipid antibody syndrome (odds ratio (OR), 8.26; 95% CI, 5.40–12.62), AT deficiency (OR, 5.15; 95% CI, 2.84–9.34) and homozygous FVL mutation (OR, 3.93; 95% CI, 2.10–7.34) influenced further treatment the most (Table 4).

### 3.4. Association between Thrombophilia and Thromboembolic Events or Pregnancy-Related Morbidity during Follow-Up

In 2429 patients with follow-up >30 days, 255 events (10.5%) occurred, comprising 142 VTE (5.8%), 91 arterial thrombosis (3.7%) and 22 pregnancy-related morbidities (1.4% of women with follow-up) during the follow-up period, which corresponds to an incidence rate per 100 person years of 1.4 (95% CI, 1.2–1.7), 0.89 (95% CI, 0.72–1.1) and 0.21 (95% CI, 0.13–0.32), respectively. Distributions of clinical characteristics and prevalence of thrombophilia of patients with and without follow-up for>30 days are represented in Appendix A. 

Compared with patients without thrombophilia, patients with high-risk thrombophilia had a higher risk of VTE (adjusted HR, 2.55; 95% CI, 1.49–4.35) during follow-up, and patients with antiphospholipid antibody syndrome a higher risk of VTE (adjusted HR, adjusted HR, 2.50; 95% CI, 1.20–5.19) and pregnancy-related morbidity (HR, 3.86; 95% CI, 1.07–13.97) during follow-up (Figure 3, Table 5). Low-risk thrombophilia was not associated with venous or arterial thrombosis or pregnancy-associated morbidity during follow-up. None of the thrombophilias was associated with arterial thrombosis during follow-up (Figure 3, Table 5). Notably, hereditary high-risk thrombophilia and antiphospholipid antibody syndrome were not only associated with a subsequent VTE during follow-up in the entire study cohort, but also with recurrent VTE, after the exclusion of patients with arterial thrombosis and pregnancy-related morbidity and asymptomatic patients (HR, 2.00; 95% CI, 1.12- 3.57; and HR, 2.82; 95% CI, 1.28–6.29, respectively). 

## 4. Discussion

In this large retrospective cohort study of 3550 patients performed at a single tertiary hematology department, we evaluated the impact of thrombophilia work-up and its result on treatment decisions, as well as the association between thrombophilia and thromboembolic events and pregnancy-related morbidity, in a real-world setting. In 86% of patients in this cohort, the result of the thrombophilia work-up did not lead to a change in treatment decision, while, in 5% of the cohort, patients’ anticoagulation was withheld inappropriately because of a negative thrombophilia work-up. Positive work-ups for hereditary high-risk thrombophilia and antiphospholipid antibody syndrome had significantly more influence on treatment decisions compared to low-risk thrombophilia, and patients with these thrombophilias were more likely to develop subsequent VTE and pregnancy-related morbidity during follow-up.

Considering that the aim of our study was to demonstrate the utility of thrombophilia work-up in a real-world setting in a tertiary care center, the cohort reflected the expected clinical characteristics of patients pre-selected by general practitioners or other specialists. Therefore, the cohort comprised young patients with few co-morbidities or major risk factors for thromboembolism. The prevalence of thrombophilia was in line with previously reported cohorts of thrombophilia work-up in Europe [21,46,47,48]. Data from our center show the limited clinical utility of thrombophilia work-up in reference to the length of anticoagulation in already pre-selected younger patients, which is in line with the findings of a few other studies in smaller, combined inpatient and outpatient VTE or arterial thrombosis cohorts from a tertiary non-hematologic care center [49,50] and tertiary hematology center [6]. Furthermore, an adverse outcome of the work-up could be demonstrated in terms of the withholding of indicated anticoagulant treatment in 5% of the cohort patients due to a negative thrombophilia work-up. Although a high-risk thrombophilia was only found in 10% of the cohort patients, it contributed to the treatment decision in 4.3% of cohort patients, providing a greater impact compared to low-risk hereditary thrombophilia. Besides continuous training to interpret the results of thrombophilia work-up, better-defined predictive factors for hereditary and acquired high-risk thrombophilia are needed to increase the utility of the work-up. 

The presence of high-risk hereditary thrombophilia and antiphospholipid antibody syndrome was independently associated with subsequent and recurrent VTE and antiphospholipid antibody syndrome with pregnancy-related morbidity after the work-up. Although the thrombophilia risk for first VTE is well established [51], data on the recurrence of VTE are less clear. Multiple reports indicate no association of a positive thrombophilia result with the recurrence of VTE, mostly in low-risk hereditary thrombophilia [9,10,12], and heterogeneous results on recurrent pregnancy morbidity [52,53,54] with the exception of antiphospholipid antibody syndrome [55]. However, data on hereditary high-risk thrombophilia are limited and studies including the whole panel of thrombophilia are sparse. To our knowledge, this is the first study reporting on new and recurrent thrombotic events and pregnancy-related morbidity after the work-up, including all types of main thrombophilia and indications of testing from real-world practice. Therefore, it seems that despite the treatment change due to high-risk thrombophilia, it still has an impact on the clinical course of VTE and pregnancy-related morbidities and its clinical management should be further investigated. 

Our study has several limitations. Firstly, the retrospective study design may have introduced selection and information bias due to the possible misclassification of primary outcomes. Nevertheless, because of the introduction of a structured reporting system and testing pattern in our center before 2010, and cross-evaluation of the data by two individuals, missing values and random errors were limited. Secondly, we could not consistently retrieve information on death, major bleeding and anticoagulation status at the time of subsequent thrombotic events and pregnancy-related morbidity due to insufficient documentation of these events and regulatory restrictions to search civil registries for death. Therefore, our findings do not allow the balancing of the risks and benefits of anticoagulant interventions and treatment decisions, and, moreover, analyses could not be adjusted for death as a competing event, which may lead to biased outcome rates. Thirdly, the definition of high-risk thrombophilia is not well established and may be debatable, especially in the context of pregnancy-related morbidity. Nevertheless, the study gives a new, comprehensive insight into the impact and outcome of thrombophilia work-up and its result. Fourthly, a very small proportion of patients with PS type II deficiency might have been missed, because no systemic measurement of PS activity was performed. Lastly, we evaluated the impact of thrombophilia on prophylactic and therapeutic anticoagulation treatment and did not consider other positive aspects of the work-up, such as a change in anticoagulant in reference to antiphospholipid antibody syndrome or any other high-risk thrombophilia, stronger motivation for lifestyle modification, avoidance of estrogen-based treatments or positive psychological effects of testing negative. Therefore, no conclusions can be made about the overall benefit–risk balance of the testing. 

In conclusion, we found that the benefit of thrombophilia testing is limited in already pre-selected outpatients and has some adverse effects on the clinician’s management of anticoagulation in all types of index thrombotic events and pregnancy-related morbidity. Better selection criteria to identify patients who may benefit from testing for hereditary and acquired high-risk thrombophilia are needed to improve the diagnostic and therapeutic yield of thrombophilia work-up and reduce the risk of inappropriate management decisions based on negative tests, the high costs of the investigations and the unfavorable impact on the psychological status of patients due to the results of unnecessary tests. Therefore, the clinical utility of the current selection criteria and the strongest factors associated with treatment should be investigated in order to establish better clinically oriented testing guidelines for thrombophilia work-up.

## Figures and Tables

**Figure 1 jcm-11-04188-f001:**
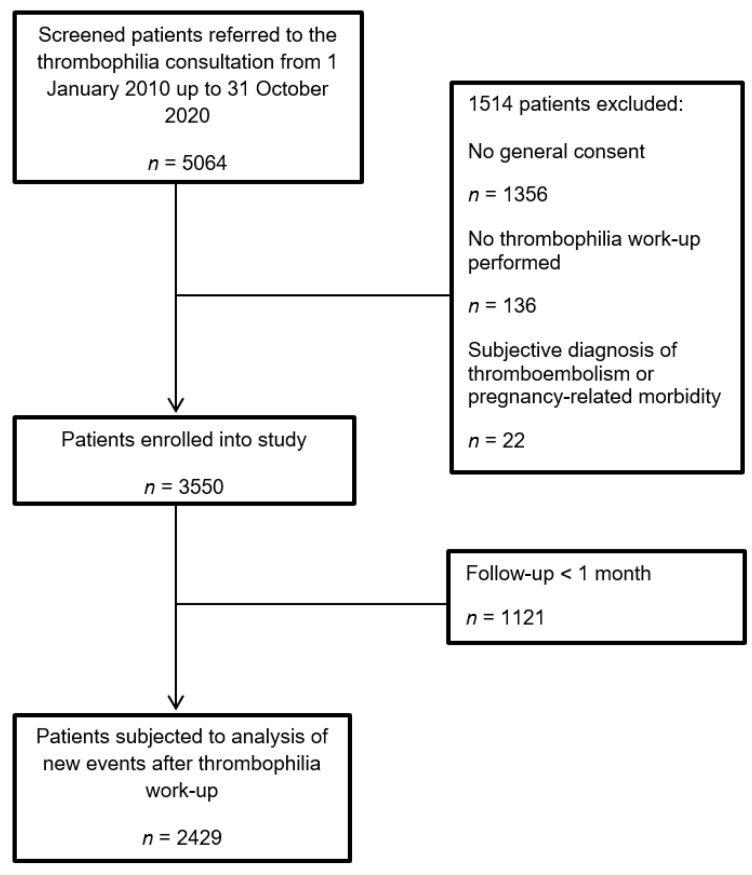
Flow diagram of patients.

**Figure 2 jcm-11-04188-f002:**
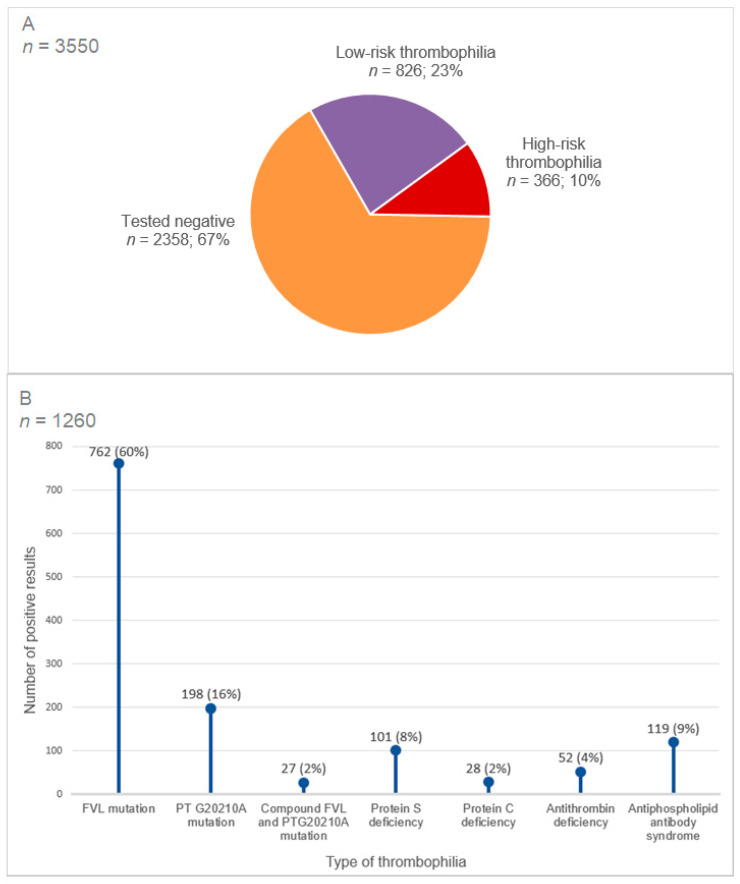
Prevalence of thrombophilia in the cohort. (**A**) Prevalence of high-risk and low-risk thrombophilia in the cohort. (**B**) Type of thrombophilia in the cohort. Abbreviations: FVL, factor V Leiden; PT, prothrombin. Testing was not performed or missing for presence of FVL mutation (6%), PT G20210A mutation (13%), antithrombin deficiency (20%), protein C deficiency (30%), protein S deficiency (29%) and antiphospholipid antibody syndrome (11%). Low-risk thrombophilia comprises heterozygous factor V Leiden or heterozygous prothrombin 20210G>A mutation. High-risk thrombophilia comprises homozygous factor V Leiden mutation, homozygous prothrombin 20210G>A mutation, antithrombin < 70%, protein C < 69% and protein S < 59%, antiphospholipid antibody syndrome and compound thrombophilias.

**Figure 3 jcm-11-04188-f003:**
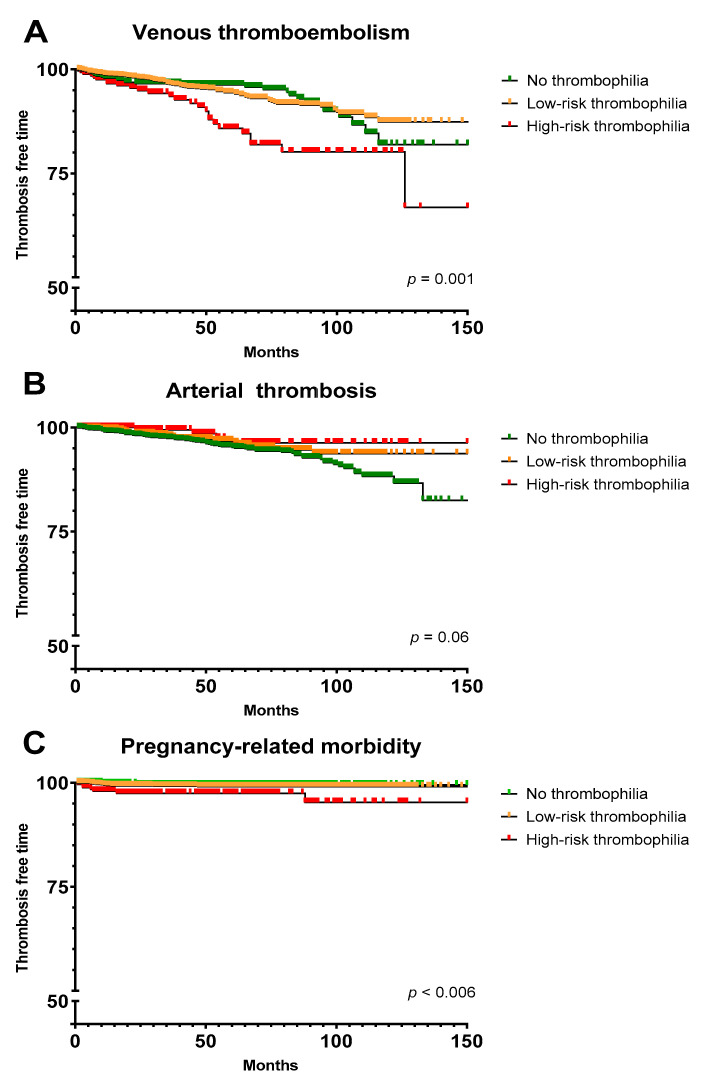
Kaplan–Meier survival curves in patients with no thrombophilia, low-risk thrombophilia and high-risk thrombophilia. (**A**) Patients with subsequent venous thromboembolism. (**B**) Patients with subsequent arterial thrombosis. (**C**) Women with subsequent pregnancy-related morbidity. Due to small sample size, a modification of y-axis scale was applied for presentation purposes. Low-risk thrombophilia comprises heterozygous factor V Leiden or heterozygous prothrombin 20210G>A mutation. High-risk thrombophilia comprises homozygous factor V Leiden mutation, homozygous prothrombin 20210G>A mutation, antithrombin < 70%, protein C < 69% and protein S < 59%, antiphospholipid antibody syndrome and compound thrombophilias.

**Table 1 jcm-11-04188-t001:** Classification of thrombophilia result influence on treatment decisions.

	No influence on treatment	Anticoagulation therapy or prophylaxis should have been initiated irrespective of thrombophilia testing result
**Positive or potential positive influence**	Appropriate decision	Decision to extend, intensify or initiate any type of anticoagulation based on a thrombophilia testing result
	Result not considered/overlooked	Thrombophilia testing result not considered in treatment decision, although it should have been
**Negative influence**	Decision to undertreat	Decision to withhold or not initiate any type of anticoagulation because thrombophilia testing result was not in accordance with guidelines
	Decision to overtreat	Decision to extend or initiate any type of anticoagulation based on a thrombophilia testing result not in accordance with guidelines

**Table 2 jcm-11-04188-t002:** Clinical characteristics of the patients included in the study in accordance with thrombophilia work-up result.

Characteristic	Tested Patients *n*= 3550	Negative Work-Up *n* = 2358 (66)	Positive Work-Up *n* = 1192 (34)	*p*-Value
Age, year, mean (±SD) *	42 (15)	43 (15)	39 (15)	
**Sex, *n* (%)**				<0.001
Female	2118 (60)	1358 (58)	760 (64)	
**Indication for consulting, *n* (%)**				<0.001
Arterial thrombosis	583 (16)	455 (19)	128 (11)	
VTE	2343 (66)	1587 (67)	756 (63)	
Pregnancy-related morbidity	120 (3.4)	59 (2.5)	61 (5.1)	
Asymptomatic patients	504 (14)	257 (11)	247 (21)	
**Provoking factors of VTE ^†^, *n* (%)**				0.026
Unprovoked VTE	683 (19)	460 (20)	223 (19)	
Provoked VTE, minor risk factor	1242 (35)	821 (35)	421 (35)	
Provoked VTE, major risk factor	415 (12)	303 (13)	112 (9.4)	
**Referral for recurrent VTE ^†^, *n* (%)**				0.009
Yes	571 (16)	365 (15)	206 (17)	
**Number of co-morbidities *, *n* (%)**				<0.001
0	1999 (56)	1231 (52)	768 (64)	
1	814 (23)	569 (24)	245 (21)	
2 or more	737 (21)	558 (24)	179 (15)	
**Number of risk factors for thromboembolism *, *n* (%)**				<0.001
0	1259 (35)	783 (33)	467 (40)	
1	1189 (33)	791 (34)	398 (33)	
2 or more	1102 (31)	784 (33)	318 (27)	
**Family history of VTE in first-degree relatives ^†^, *n* (%)**				<0.001
Positive	1106 (31)	643 (27)	463 (39)	
**Family history of VTE in second-degree relatives ^†^, *n* (%)**				<0.001
Positive	523 (15)	315 (13)	208 (17)	

Abbreviations: SD, standard deviation; VTE, venous thromboembolism. Categorical values are compared by x^2^ test and continuous variables by ANOVA. Risk factors include smoking, immobilization > 4 h, cancer, central intravenous catheter, infection, estrogen-based treatment, pregnancy, cancer, obesity, trauma, surgery, cancer and its medication. Co-morbidities include diabetes, arterial hypertension, liver cirrhosis, kidney failure, rheumatic diseases, depression, dyslipidemia, lung diseases, neurological disorders, cardiovascular diseases and chronic inflammatory diseases. * At time of VTE, arterial thrombosis or pregnancy-related morbidity or at time of consultation in asymptomatic patients. ^†^ Values were missing for provoking factors of VTE (0.08%), history of prior VTE at time of consultation (0.8%), family history of VTE in first-degree (1.3%) and second-degree (1.7%) relatives.

**Table 3 jcm-11-04188-t003:** Influence of thrombophilia work-up on treatment decision.

	Total	No Influence on Therapy	Positive and Potential Positive Influence		Negative Influence		*p*-Value
			Appropriate Decision	Overlooked Results	Decision to Overtreat	Decision to Undertreat	
	*n* = 3550	*n* = 3050 (85.9)	*n* = 211 (5.7)	*n* = 82 (2.2)	*n* = 11 (0.3)	*n* = 184 (5.1)	<0.001
Negative thrombophilia work-up, *n* (%)	2358 (66)	2171 (71)	0	0	1 (9.1)	181 (98)	
Hereditary low-risk thrombophilia, *n* (%)	826 (23)	675 (22)	111 (53)	25 (30)	7 (64)	3 (1.6)	
Hereditary high-risk thrombophilia, *n* (%)	247 (6.3)	157 (5.1)	50 (24)	36 (44)	2 (18)	0	
Antiphospholipid antibody syndrome, *n* (%)	119 (3.4)	47 (1.5)	50 (24)	21 (26)	1 (9.1)	0	

Categorical variables are compared by x^2^ test. Twelve work-ups could not be categorized due to unclear statement on treatment decision in the clinical report. Low-risk thrombophilia is defined by the presence of heterozygous factor V Leiden, heterozygous prothrombin 20210G>A mutation; high-risk thrombophilia comprises homozygous factor V Leiden mutation, homozygous prothrombin 20210G>A mutation, antithrombin < 70%, protein C < 69%, protein S < 59% and compound thrombophilias.

**Table 4 jcm-11-04188-t004:** Influence of the type of thrombophilia on therapy.

Type of Thrombophilia	OR (95% CI)
Heterozygous factor V Leiden mutation	1 (ref)
Antiphospholipid antibody syndrome	8.26 (5.40–12.62)
Antithrombin < 70%	5.15 (2.84–9.34)
Homozygous factor V Leiden mutation	3.93 (2.10–7.34)
Protein S < 59%	1.99 (1.21–3.27)
Heterozygous prothrombin 20210G>A mutation	1.89 (1.24–2.90)
Homozygous prothrombin 20210G>A mutation	2.79 (0.25–31.07)
Protein C < 69%	2.17 (0.88–5.33)

Abbreviations: OR, odds ratio. ORs are calculated by univariable logistic regression, using heterozygous factor V Leiden mutation as a reference due to its lowest influence on treatment decision.

**Table 5 jcm-11-04188-t005:** Cause-specific hazard ratios for subsequent events after thrombophilia testing during follow-up according to thrombophilia status.

	Crude HR (95% CI)	Adjusted HR (95% CI)
Venous thromboembolism		
Negative work-up	1 (ref.)	1 (ref.)
Hereditary low-risk thrombophilia	1.02 (0.66–1.56)	1.08 (0.70–1.67)
Hereditary high-risk thrombophilia	1.99 (1.18–3.36)	2.55 (1.49–4.35)
APS	2.33 (1.13–4.84)	2.50 (1.20–5.19)
Arterial thrombosis		
Negative work-up	1 (ref.)	1 (ref.)
Hereditary low-risk thrombophilia	0.69 (0.40–1.18)	0.86 (0.50–1.50)
Hereditary high-risk thrombophilia	0.27 (0.07– 1.11)	0.38 (0.09–1.58)
APS	0.70 (0.17–2.85)	0.82 (0.20–3.37)
Pregnancy-related morbidity		
Negative work-up	1 (ref.)	1 (ref.)
Hereditary low-risk thrombophilia	0.76 (0.21–2.69)	0.57 (0.16–2.04)
Hereditary high-risk thrombophilia	3.23 (1.04–10.00)	1.93 (0.62–6.05)
APS	4.49 (1.27–15.96)	3.86 (1.07–13.97)

Abbreviations: APS, antiphospholipid antibody syndrome; HR, hazard ratio; CI, confidence interval. Low-risk thrombophilia is defined by the presence of heterozygous factor V Leiden or heterozygous prothrombin 20210G>A mutation; high-risk thrombophilia comprises homozygous factor V Leiden, homozygous prothrombin 20210G>A mutation, antithrombin <70%, protein C <69%, protein S <59% and compound heterozygous factor V Leiden and prothrombin 20210G>A mutations. Adjusted cause-specific HRs were calculated in a multivariable Cox model, including age >50, male gender, history of prior VTE at time of consultation, smoking, diabetes mellitus, obesity (body mass index >30 kg/m^2^), arterial hypertension, kidney failure, dyslipidemia, depression, chronic inflammatory disease and active cancer. No signs of non-proportional hazards were found.

## Data Availability

There were no publicly archived datasets analyzed or generated during the study.

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
