# Peer review of "Thrombophilia Impact on Treatment Decisions, Subsequent Venous or Arterial Thrombosis and Pregnancy-Related Morbidity: A Retrospective Single-Center Cohort Study"

_jcm, 2022, doi:10.3390/jcm11144188_

Round 1

Reviewer 1 Report

This is a nicely written and comprehensive manuscript of thrombophilia impact on treatment decisions. The authors do a solid job.  I have no further comments or recommendations.  I recommend publishing this manuscript.

Author Response

We thank this reviewer for the careful review and the positive global assessment of our work and clear and conclusive statement.

Reviewer 2 Report

It is a valuable research, dedicated to a problem currently of great interest worldwide. It is a single center retrospective  (median monitoring duration 49 months) large cohort (3550/2429 enrolled/monitored  respectively) aiming  at  finding the correct response  to the question of the usefulness on therapy decision making of largely performed  thrombophilia work-up. The objectives, material, methods and results are clearly presented; the analyze of data regarding  the useful negative/positive influence of the work-up are supported by a rigorous statistical evaluation. There are mentioned all the limitations of the study. The conclusions are argued with the concrete data expressed by the achieved results.

Recommendations for correction:

 - page 1/row 20,21,22-to add beside low and high risk thrombophilia also the word “factor”

- page 6 - in connection with Table1 I would introduce in the text  beside the dispositional thrombophilic factors also some explanatory words regarding the impact of the  large list of expositional factors (comorbidities, medical risk factors, aso )

- page 13 -in the conclusions I would add also  the high costs of the investigations and  the unfavourable impact on the psychological status of patients due to the  results of unnecessary investigations; but the main reason for a better selection of the criteria to identify persons for thrombophilia work-up will remain the benefit for the therapy strategy and best management of the  patient’s care.

-some minor corrections in the references-3,21,37

Author Response

We thank this reviewer for the careful review and the positive global assessment of our work.

Recommendations for correction:

Point 1

- page 1/row 20,21,22-to add beside low and high risk thrombophilia also the word “factor”

Answer point 1: Thank you very much for this remark. Due to strict world limitation in an abstract section, we decided to omit this word. Although we investigate many thrombophilic factors, we focus on thrombophilia, especially low-risk and high-risk thrombophilia, which is a general term with reference to laboratory thrombophilia work-up. We hope the reviewer will agree with this decision. 

Point 2

- page 6 - in connection with Table1 I would introduce in the text  beside the dispositional thrombophilic factors also some explanatory words regarding the impact of the  large list of expositional factors (comorbidities, medical risk factors, aso )

Answer point 2: We thank the reviewer for this remark. We included further explanation of these factors in the text accordingly: „Co-morbidities and risk factors of the cohort patients are presented in Supplemental Table 1, comprising mostly arterial hypertension (n = 578, 16%), dyslipidemia (n = 405, 11%) as risk factors for arterial thrombosis, and immobilization > 4 hours (n = 743, 21%) and estrogen-based medication (n = 706, 20%), representing minor risk factors for VTE. The major risk factors for VTE were found only in 415 (12%) cohort patients.

Point 3

- page 13 -in the conclusions I would add also  the high costs of the investigations and  the unfavourable impact on the psychological status of patients due to the  results of unnecessary investigations; but the main reason for a better selection of the criteria to identify persons for thrombophilia work-up will remain the benefit for the therapy strategy and best management of the  patient’s care.

Answer point 3: Thank you very much for pointing this out, this is indeed very important. We added this suggestion in our conclusion accordingly: In conclusion, we found that the benefit of thrombophilia testing is limited in already preselected outpatients and has some adverse effect on the clinician's management of anticoagulation in all types of index thrombotic events and pregnancy-related morbidity. Better selection criteria to identify patients who may benefit from testing for hereditary and acquired high-risk thrombophilia are needed to improve the diagnostic and therapeutic yield of thrombophilia work-up and reduce the risk of inappropriate management decisions based on negative tests, high costs of the investigations and the unfavourable impact on the psychological status of patients due to the results of unnecessary tests. Therefore, the clinical utility of current selection criteria and the strongest factors associated with treatment should be investigated in order to establish better clinically oriented testing guidelines for thrombophilia work-up.   

Point 4

-some minor corrections in the references-3,21,37

Answer point 4: We corrected the references and used standard IM abbrevations for journals.

Reviewer 3 Report

The manuscript by Vrotniakaite-Bajerciene et al., is a retrospective study involving 3550 patients referred for thrombophilia consultation at a single centre in Bern between 2010 and 2020. The aim was to investigate the impact of thrombophilia testing on management decisions and thromboembolic clinical events. The study concluded that thrombophilia testing had limited impact on clinical decisions. This work also found that high-risk thrombophilia was associated with morbidity in pregnancy and thromboembolism.

The manuscript is well-written and the results are presented clearly. This work is important given the paucity of ‘real world’ data informing clinical decisions about thrombophilia testing. Overall, this study re-enforces the existing ideas that testing does not lead to significant outcomes in terms of VTE and may in some cases contribute to misinterpretation of testing results in the clinic.

Comments

One of the limitations of the study is the lack of data about anticoagulation at the time of subsequent thrombotic events. Since the duration of anticoagulation is an important consideration, did the data show a change in decision making regarding duration of anticoagulation following thrombophilia testing? Did testing influence the type of anticoagulant used?

The conclusion mentions “better selection criteria to identify patients who benefit from testing for hereditary and acquired high-risk thrombophilia are needed”. Could the authors please provide some ideas about these new criteria that could be useful in clinical practice?

Is the data strong enough to recommend against thrombophilia testing in general? If not, what additional studies would be required/recommended for a more definitive conclusion?

Minor point

Please use standard IM abbreviations for Journals' names in the references section.

Author Response

We thank this reviewer for the careful review and the positive global assessment of our work.

Point 1

One of the limitations of the study is the lack of data about anticoagulation at the time of subsequent thrombotic events. Since the duration of anticoagulation is an important consideration, did the data show a change in decision making regarding duration of anticoagulation following thrombophilia testing? Did testing influence the type of anticoagulant used?

Answer point 1: Thank you for this important remark.  

Our primary endpoint generally investigated the impact of thrombophilia-work up on the anticoagulation treatment and its extension, initiation or termination. We wanted to investigate the impact on the exact duration of anticoagulation as well and initially classified this variable in the clinical report form as long-term; anticoagulation 6-12 months with secondary prophylaxis; anticoagulation for 3-6 months with secondary prophylaxis; primary prophylaxis and wanted to include this important variable in the analysis, especially with reference to secondary study outcome.

 However, as anticoagulation status could not be retrieved at the end of the follow-up in most of the patients and double-checked, if the suggestion of the center was implemented, we decided not to perform further investigations and acknowledge this limitation in the limitation section of the study. Although the decision most probably was implemented, as referring doctors usually were asking this question, we still were unable to prove it. Secondly, we always suggested re-evaluation of patients and their anticoagulation annually considering the bleeding risk by the referring doctor. Consequently, the anticoagulation beyond this time point may have been changed. Therefore, we decided not to interpret this variable on the exact length of anticoagulation and focused on the time point of thrombophilia work-up, if it had any influence on further anticoagulation of the patients by the specialists.  

Regarding the type of anticoagulant and its change because of thrombophilia result, we could not retrieve this data accurately. Our center usually gives a clear statement regarding the length of anticoagulation, however suggests and discusses all possible choices of anticoagulation for the referring doctors, with an exception for VKA implementation in terms of high-risk antiphospholipid antibody syndrome. Consequently, no clear conclusion could have been drawn regarding this question and we acknowledge this in our limitation section as well.

 Point 2

The conclusion mentions “better selection criteria to identify patients who benefit from testing for hereditary and acquired high-risk thrombophilia are needed”. Could the authors please provide some ideas about these new criteria that could be useful in clinical practice?

Answer point 2: We completely agree with the importance of the further exploration of this topic and thank the reviewer for pointing this out. We indeed investigated this question and calculated the odds ratio of all clinical characteristics associated with treatment decisions and compared this to current selection criteria of testing and discussed this in another manuscript. We recently submitted this publication and it is in a peer review process. As it is a big discussion topic with much novel data, we decided to write a separate manuscript on this topic in order to present the data more clearly, thorough and precisely. We included this following continuum in the conclusion as follows: “In conclusion, we found that the benefit of thrombophilia testing is limited in already preselected outpatients and has some adverse effects on the clinician's management of anticoagulation in all types of index thrombotic events and pregnancy-related morbidity. Better selection criteria to identify patients who benefit from testing for hereditary and acquired high-risk thrombophilia are needed to improve the diagnostic and therapeutic yield of thrombophilia work-up and reduce the risk of inappropriate management decisions based on negative tests, high costs of the investigations and the unfavourable impact on the psychological status of patients due to the results of unnecessary tests. Therefore, the clinical utility of current selection criteria and the strongest factors associated with treatment should be investigated in order to establish better clinically oriented testing guidelines for thrombophilia work-up. 

Point 3

Is the data strong enough to recommend against thrombophilia testing in general? If not, what additional studies would be required/recommended for a more definitive conclusion?

Answer point 3: Thank you for pointing this out. We indeed discussed this issue extensively among the authors and the doctors of our centre. We believe the data are strong enough to suggest the limited usefulness of the testing, however not completely stop the testing, as high-risk thrombophilia has significantly more impact compared to low-risk thrombophilia. Therefore, we conclude that better selection criteria to increase the diagnostic yield of the work-up for high-risk thrombophilia are needed. 

We decided to discuss this topic more in details in our conclusion according to your comment: “In conclusion, we found that the benefit of thrombophilia testing is limited in already preselected outpatients and has some adverse effect on the clinician's management of anticoagulation in all types of index thrombotic events and pregnancy-related morbidity. Better selection criteria to identify patients who benefit from testing for hereditary and acquired high-risk thrombophilia are needed to improve the diagnostic and therapeutic yield of thrombophilia work-up and reduce the risk of inappropriate management decisions based on negative tests, high costs of the investigations and the unfavourable impact on the psychological status of patients due to the results of unnecessary tests. Therefore, the clinical utility of current selection criteria and the strongest factors associated with treatment should be investigated in order to establish better clinically oriented testing guidelines for thrombophilia work-up.   

Point 4

Minor point

Please use standard IM abbreviations for Journals' names in the references section.

Answer point 4: We corrected the references and used standard IM abbrevations for journals.